# Comparative Analysis of Free-Circulating and Vesicle-Associated Plasma microRNAs of Healthy Controls and Early-Stage Lung Cancer Patients

**DOI:** 10.3390/pharmaceutics14102029

**Published:** 2022-09-23

**Authors:** Luigi Pasini, Ivan Vannini, Paola Ulivi, Michela Tebaldi, Elisabetta Petracci, Francesco Fabbri, Franco Stella, Milena Urbini

**Affiliations:** 1Biosciences Laboratory, IRCCS Istituto Romagnolo per lo Studio dei Tumori (IRST) “Dino Amadori”, 47014 Meldola, Italy; 2Unit of Biostatistics and Clinical Trials, IRCCS Istituto Romagnolo per lo Studio dei Tumori (IRST) “Dino Amadori”, 47014 Meldola, Italy; 3Division of Thoracic Surgery, G.B. Morgagni-L. Pierantoni Hospital, 47121 Forlì, Italy

**Keywords:** extracellular vesicles, cell-free RNA, liquid biopsy, lung cancer, NSCLC, plasma, miRNA sequencing

## Abstract

In recent years, circulating extracellular miRNAs have emerged as a useful tool for the molecular characterization and study of tumors’ biological functions. However, the high heterogeneity in sample processing, isolation of circulating fraction, RNA extraction, and sequencing hamper the reproducibility and the introduction of these biomarkers in clinical practice. In this paper, we compare the content and the performance of miRNA sequencing in plasma-derived samples processed with different isolation protocols. We tested three different fractions of miRNA from healthy-donor human blood: whole plasma (WP), free-circulating (FC) and EV-associated, isolated by either column (ccEV) or size exclusion chromatography (secEV) miRNAs. An additional cohort of 18 lung cancer patients was analyzed. Protein profiles of ccEV and secEV were compared and miRNA expression profiles were assessed through sequencing. Slight differences were found between ccEV and secEV expressions of typical EV markers. Conversely, sequencing performance and the mirnome profile varied between RNA extracted using different isolation methods. Sequencing performance was better in FC samples. Higher varieties of miRNAs were identified in WP and FC with respect to ccEV and secEV. Analysis of free-circulating and EV-associated miRNA profiles in lung cancer patients demonstrated the reliability of the biomarkers identifiable on plasma with these approaches.

## 1. Introduction

In recent years, liquid biopsy analysis has revolutionized the ability to identify, measure, and monitor biomarkers useful in early cancer detection and therapeutic decisions [1,2]. These molecular biomarkers can be derived from blood, urine, and other biofluids. However, most clinical tests are based on plasma analysis. In particular, analysis of plasma-derived biomarkers (specifically cell-free DNA) has been accepted in the clinical practice for non-small cell lung cancer (NSCLC) treatment decision-making [3,4]. Although the current clinical diagnosis of NSCLC has been based on mutational analysis of recurrent driver oncogenes, sequencing of cellular and extracellular miRNAs has emerged as an important source of information, useful for the molecular characterization and study of tumors’ biological functions [5]. In fact, miRNAs are short RNA stretches of about 22 nt that can regulate the expression of many target genes involved in tumor development and progression [6,7]. Importantly, when released from the cell, miRNAs can stay stable for several hours in the bloodstream before reaching their target organ [6]. Circulating miRNAs can be indeed secreted either in complexes with specific RNA-binding proteins or transported by extracellular vesicles (EVs). The fact that both free-circulating miRNAs (fcmiRNAs) and EV-associated miRNAs can be easily detected and analyzed from body fluids has raised new hopes for making accurate diagnoses of cancer by routine liquid biopsy [8,9]. However, several studies have evaluated the expression of miRNAs in human plasma, but none of those miRNAs found to be candidate oncogenic biomarkers have actually been used in the clinical practice of cancer, yet [7]. This problem may be partly due to the heterogeneity of methods used for isolation and characterization of circulating miRNAs causing difficulty in the identification of a reproducible miRNA signature for disease monitoring [10]. This methodological heterogeneity ranges from sample collection (tube types, centrifugation steps, plasma volumes, etc.), to the fractions of circulating miRNAs being considered (free-circulating versus vesicular miRNAs, or total plasma miRNAs), and to the methods used for library preparation and sequencing (targeted miRNA detection, small RNA or total RNA sequencing). In particular, thanks to the tremendous lowering of market costs, deep RNA sequencing has become the gold standard for oncogenic miRNA screening on large scale, but optimization of the technique is still required, especially when dealing with liquid biopsy samples [11]. Here, we aim to provide useful information on the performance of miRNA sequencing applied to plasma-derived samples, highlighting the strengths and limitations of this approach, and compare the results obtained from separating and sequencing three different fractions of miRNA from human blood: whole plasma (WP) miRNA, fcmiRNA, and EV-associated miRNA. Specifically, we tested side-by-side the isolation of blood EVs with either column chromatography (ccEV) or with size exclusion chromatography (secEV); two technologies that are commercially available. Finally, we verified our data on a small cohort of early-stage NSCLC patients.

## 2. Materials and Methods

### 2.1. Sample Collection

Blood samples (20 mL) from three healthy volunteering donors and 18 patients with early-stage NSCLC were collected in two Cell-Free DNA BCT Streck tubes (Streck Corporate), and plasma was isolated within two hours of blood withdrawal. Blood was first centrifuged at 1600× *g* for 10 min, followed by a second centrifugation of the plasma supernatant at 2000× *g* for 10 min to ensure complete removal of any cellular debris. The final supernatant was stored at −80 °C until use. In total, 6 mL of plasma were collected to allow the comparison of the different extraction methods (2 mL of plasma was used for each extraction method). The study was performed in accordance with the principles of the Declaration of Helsinki. The collection of blood samples from NSCLC patients and healthy donors was approved by the local Ethics Committee of IRST-IRCCS, after obtaining patient informed consent (Prot. No. IRST B078; approval date 18 January 2018). For the healthy volunteers, consent was waived by the Ethics Committee. Clinical characteristics of the NSCLC patients are reported in Table 1.

### 2.2. EV Isolation and RNA Extraction

Spin column chromatography (CC) by silicon carbide resin separation matrix was used to extract both total plasma RNA (all sizes of circulating and vesicular RNA) and the separated fraction of vesicular and cell-free circulating RNA. Specifically, whole plasma RNA (WP-RNA) was extracted by using the Plasma/Serum RNA Purification Midi Kit (Cat. 56100, Norgen Biotek Corp, Thorold, ON, Canada). Parallel extraction of EV-RNA (ccEV-RNA) and free-circulating RNA (FC-RNA) from the same aliquot of plasma were isolated by using the Plasma/Serum Exosome and Free-Circulating RNA Isolation Midi Kit (Cat. 59600, Norgen Biotek Corp, Thorold, ON, Canada). Plasma EVs were also isolated from plasma through size exclusion chromatography (SEC) columns of polysaccharide resin by using the qEV 70 columns (IZON), and the corresponding vesicular RNA (secEV-RNA) was extracted by using the Plasma/Serum RNA Purification Midi Kit (Cat. 56100, Norgen Biotek Corp, Thorold, ON, Canada). For each EV and RNA extraction method tested, 2 mL of plasma was used. 

### 2.3. Nanoparticle Analysis

Isolated EVs were first characterized for size, concentration, and polydispersity data by using the NanoSight NS300 tracking system (Malvern Instruments Limited, Cambridge, UK). Samples were diluted in PBS to a final volume of 1 mL and concentration settings were set according to the manufacturer’s software manual, within the particles/frame range of 20–120 and the total track to valid track ratio of less than 5 (NanoSight NS300 User Manual, MAN0541-01-EN-00, 2017). The camera level was increased until all particles were distinctly visible but not exceeding the particle signal saturation of over 20%. For each measurement, three videos of 30 s were captured with a cell temperature of 25 °C and a syringe speed of 40 µL/s. Videos were analyzed by the NanoSight Software NTA 3.1. Flow cytometric analyses of isolated EVs were conducted by using the MACSPlex Exosome Kit, human (Miltenyi Biotec B.V. & CO. KG, Bergisch Gladbach, Germany) that allows the detection of 37 distinct EV-surface epitopes and two isotype controls, which can be distinguished by different fluorescence intensities by flow cytometry. Samples were diluted with MACSPlex buffer (MPB) to a final volume of 120 µL, and 15 µL of MACSPlex Exosome Capture Beads and 5 uL of each of MACSPlex Exosome Detection Reagent antibodies (CD9, CD63, CD81) were added to each well. The samples were then incubated on an orbital shaker for 1 h at 450 rpm at room temperature protected from light. Beads were washed with 200 µL of MPB and centrifuged at 3000× *g* for 5 min and removed 500 μL supernatant. Next, plates were washed by adding 500 µL MPB and centrifuged at 3000× *g* for 5 min and removed 500 μL supernatant. This step was followed by another washing step with 500 µL of MPB, incubation on an orbital shaker at 450 rpm protected from light for 15 min at room temperature, and centrifugation at 3000× *g* for 5 min to remove the supernatant. Flow cytometric analysis was performed with a BD FACSVantage™ cytofluorimetric (BD Biosciences, Franklin Lakes, NJ, USA). Approximately 10,000 events were recorded per sample. Median fluorescence intensity (MFI) for all 39 capture bead subsets was background corrected by subtracting respective MFI values from matched non-EV buffer or media controls that were treated exactly the same as EV-containing samples. GraphPadPrism 6 (GraphPadPrism Software, La Jolla, CA, USA) was used to analyze data and assemble figures.

### 2.4. Protein Analysis of EV lysates

Proteins were precipitated from the EV eluate following the Liopis et al. protocol [12], with some modifications. The EV eluates were mixed with 1 volume of acid-phenol:chloroform to separate the lower organic phase, containing proteins, from the upper aqueous phase. Proteins were precipitated with ice-cold acetone (2.5 volumes) for 30 min at −20 °C. Samples were then centrifuged for 5 min at 10,000× *g*, the supernatant discarded, and the pellet washed with 1 mL of ice-cold acetone first and subsequently with 1 mL of ice-cold 95% ethanol. The pellet was resuspended in 30 uL of protein lysis buffer (20 mM EDTA, 140 mM NaCl, 5% SDS, 100 mM Tris Ph 8, 1 mM Na Orthovanadate, 2 mM PMSF) by incubating the in a heat block at 56 °C for 10 min. Residual material was eliminated from the protein supernatant by centrifuge at full speed for 1 min. MCF7 cells were lysed with the same EV lysis buffer, sonicated, and used as control. 20 µg of total protein of each EV lysate sample were separated by SDS-PAGE on a 10% acrylamide (10% Criterion TBE Polyacrylamide Midi Gel, Bio-Rad, Hercules, CA, USA) and transferred onto a nitrocellulose membrane (GE Healthcare, Freiburg, Germany). The membranes were blocked overnight with 5% milk at 4 °C and incubated with primary antibodies against CD63 (1:1000, Santa Cruz Biotechnology Cat. sc5275), Alix (1:500, Cell Signaling Cat. 2171), HSP70 (1:1000, Cell Signaling Cat. 4876), and Actin (1:2000, Sigma-Aldrich Cat. A2066). After washing, the blots were incubated with HRP-conjugated anti-mouse IgG secondary antibody (1:5000, Santa Cruz Biotechnology Cat. sc-2005) or anti-rabbit IgG secondary antibody (1:5000, Bethyl Laboratories Cat. A120-101P) for 1 h at RT. Protein expression was visualized using the Bio-Rad ChemiDoc MP imaging system.

### 2.5. microRNA Profiling

The quality of extracted RNA was checked using RNA6000 pico chips on a bioanalyzer 2100 instrument (Agilent Technologies, Milan, Italy) and 5 uL of RNA were used for library preparation using Qiaseq miRNA library kit (Qiagen, Milan, Italy). This kit incorporates UMI (unique molecular identifier) during library synthesis, which allows the counting of the initial small RNA molecules present in the starting material, reducing PCR amplification bias. Libraries were prepared following the manufacturer’s instructions with some adaptation for low RNA inputs. In particular, 3′ and 5′ ligation adapters and RT primers were diluted at respectively 1:10 and 1:5 to reduce adapter dimer formation (approximately around 155 bp). Final PCR cycles were increased to 22 cycles. An additional bead purification step (beads:amplicon ratio of 1.4) was added to remove unwanted small fragments (<100 bp). Final libraries were then quantified using Qubit dsDNA HS assay kit (ThermoFisher, Waltham, MA, USA) and quality checked for dimension DNA high sensitivity chips on Bioanalyzer2100 instrument (Agilent). The expected library size ranges from 160 to 200 bp, with the miRNA peak located at 178 bp. Libraries were then normalized and sequenced on Nextseq550 instrument (Illumina, San Diego, CA, USA). 

### 2.6. Bioinformatic and Statistical Analysis

A local RUN manager was used for demultiplexing. Reads were then trimmed, corrected for UMIs reduction, and aligned to mirBase v22 using the miRNA default analysis of CLC Genomics software (Qiagen). CLC output statistics were used for estimating sequencing performance. For differential expression analysis of healthy donors, miRNA with UMI raw count greater than five in at least three samples was retained. All the analyses were performed using the open-source statistical computing environment R v4.1.1 (The R Project for Statistical Computing. Available online: https://www.r-project.org/; accessed on 30 June 2022, Vienna, Austria). Data normalization (using the Trimmed Mean of M-values method) and differential expression analysis were performed using the edgeR package. Other analyses and graphical representations were conducted using the following packages: EDASeq, prcomp, stats, and gplots. The top expressed miRNA of each fraction was identified in comparison with all other plasma-circulating fractions, and miRNA with log2FC > |2| with a false discovery rate (FDR) < 0.05 were considered. In silico subsampling of the reads was used to compare sequencing results from different starting sequence depths (5, 10, 15 and 20 M). The gene targets of differential miRNA were predicted and prioritized using miRWalk (miRWalk. Available online: http://mirwalk.umm.uni-heidelberg.de; accessed on 10 June 2022) and miRTarBase (miRTarBase. Available online: https://mirtarbase.cuhk.edu.cn/; accessed on 10 June 2022), GSEA was then used for gene ontology (GO) and pathway analysis.

## 3. Results

To evaluate the most suitable method for investigating the plasma fcmiRNA content of a liquid biopsy, we decided to compare two different commercially available kits for EV isolation and subsequent analysis of miRNAs transported by EVs and free-circulating miRNAs. To this scope, we analyzed three healthy donors. From the same individual, we extracted the whole plasma (WP) RNA as well as the RNA packed into the EVs, isolated with either SEC (secEV) or column chromatography (ccEV), and the free-circulating (FC) RNA fraction (Figure 1).

Mean size and plasma concentration (particles/mL) of the EVs were quantified by the NanoSight Software NTA, and particle profile distribution was evaluated for the three healthy donors, comparing EVs extracted either with SEC (where the most enriched elution fraction was considered) or CC (Figure 2A,B). Overall, by using SEC we were able to isolate more particles (mean 6.03 × 10^10^ EV/mL) from the same amount of plasma (2 mL), compared to CC (mean 3.21 × 10^10^ EV/mL), while the average particle size was similar between the two isolation methods (median 122.4 nm, SD 7.15 nm for SEC; median 114.2 nm, SD 4.73 nm for CC). To make sure we were analyzing EVs in the range of exosomes and microvesicles, we checked the expression of some characteristic vesicular markers by western blot (the Programmed Cell Death 6 Interacting Protein, Alix; the Heat Shock Protein Family A, HSP70; the CD63 Antigen, CD63; the Actin Alpha 1, Actin). The expression of these specific EV markers was similar between ccEV and secEV (different elution fractions were considered for secEV, indicating the fraction 14–15 as the most enriched for marker expression) as compared to EV isolated with a reference EV-isolation method (NBI, nickel-based isolation) [13] (Figure 2C). We also analyzed the expression of a panel of characteristic membrane epitopes present on the EV surface, by using the MACSPlex Exosome Kit, which allows the detection of 37 distinct EV surface markers, including typical tetraspanins CD9, CD63 and CD81, by flow cytometry. Comparing vesicles extracted either with SEC or CC, we found that most of the markers tested had similar expression levels, while some specific markers were more enriched in the ccEV extracts, such as CD81, while the Platelet And Endothelial Cell Adhesion Molecule 1 (CD31) and the Glycoprotein IX Platelet (CD42a) was quantified by the mean fluorescence intensity (MFI) signal (Figure 2D and Appendix A).

### 3.1. Difference of RNA and Library Profiles Obtained by Different Extraction Methods

Regardless of the extraction method used, RNA concentration was not determinable as it was below the detection limit of common UV and fluorimetric methods. Nonetheless, samples were checked on the Bioanalyzer instrument for RNA quality. Among all samples extracted, we found an enrichment of RNA fragments ranging from 50 and 100 nt. In particular, WP extracts had a higher RNA yield compared to ccEV, secEV, and FC extracts, and a small portion of longer RNA fragments (up to 200 nt) could be detected (Appendix A). Sequencing libraries were synthesized starting from 5 µL of RNA using the Qiaseq miRNA library kit. Library profiles differed slightly between the four types of extracted RNAs (WP, secEV, ccEV and FC), with FC RNA being the one most enriched in amplified fragments of 178 bp, which matched the expected size of the library containing the inserted miRNA. On the other hand, libraries derived from secEV, ccEV and WP contained a predominant peak at 160–165 nt in addition to the miRNAs peak, indicating the incorporation of smaller RNA inserts (5–15 nt) during the library preparation (Appendix A). In all cases, we did not observe the presence of any adapter dimers (155 bp), indicating the good quality of the ligation process.

### 3.2. Sequencing Performance According to Extraction Method

An average of 22 million reads per sample was produced. Reads were then checked by the bioinformatic pipeline, and reads failing quality checks (no common sequence, no UMI, wrong insert size) were discarded. Of note, FC and ccEV sequencing produced the highest percentage of passing filter (PF) reads, with an average of 64% and 52% of PF reads, respectively. Conversely, WP and secEV showed a lower fraction of PF reads, 34% and 27%, respectively (Appendix A, Figure 3A). Reads that did not pass the filtering analysis predominantly originated from libraries containing too short inserts (<15 nt) (Appendix A). Most PF reads carried insert dimensions of about 20–23 nt (corresponding to miRNA length) in all fractions, while longer inserts were almost absent. On the other side, a higher percentage of small inserts (between 15 and 20 nt) were detected in WP, ccEV and secEV with respect to FC (Figure 3B). PF reads were then collapsed to identifiable UMI groups before proceeding with mapping on the miRBase database. An average of 522,231 unique UMI groups per sample were obtained, with ccEV and secEV showing the lower number of identifiable unique UMI groups (Appendix A, Figure 3B). Interestingly, of the identifiable unique UMI groups, approximately 78% were mapped on miRNAs in FC samples, while UMI groups obtained from ccEV, secEV and WP showed lower mapability, 27%, 20% and 32%, respectively (Figure 3C). These findings are consistent with the average number of miRNA identified by at least 1 UMI read: 533, 251, 345, and 473 in FC, ccEV, secEV, and WP, respectively (Figure 3D). Then, we assessed if we could lower the sequencing depth to 5–10 million reads per sample to reach the optimal sequencing depth for each RNA fraction without affecting the quality of miRNA detection. While an evident reduction in the global number of UMI groups could be seen, only a modest decrease in identifiable miRNA was found (ranging from 1% to 6%), supporting the robustness of the assay (Figure 3E,F, Appendix A). However, for the ccEV fraction, we do not suggest reducing the sequencing depth, since we observed a significant increase of UMI singletons (groups identified by only one UMI) that mapped on miRNAs (Figure 3G) with a consequent reduction of power for the subsequent differential expression analysis.

### 3.3. Different miRNA Expression Profiles Identified According to Extraction Method

Overall, a similarity could be found between the content of miRNAs of each RNA fraction, with an average of 119 out of 335 miRNAs (36%) shared between the four RNA extraction methods. Conversely, miRNAs specifically detected in only one RNA fraction can be found. On average, WP and FC showed a relevant proportion of miRNAs found specifically only in these fractions (respectively, 8.2 and 21.8%), while less than 1% of the total miRNAs identified were exclusively found in vesicular extracts (Figure 4A). 

In total, 935 miRNAs were detected in at least one sample of the four RNA extracts and were used for differential analysis of miRNAs. Considering the expression profiles, each fraction formed distinct clusters in unsupervised analysis, with secEV and WP clustering more closely, as compared to ccEV and FC (Figure 4B). Through supervised analysis, we identified the top differentially expressed miRNAs for each circulating fraction (Appendix A and Figure 4C). All these miRNAs were reported in the literature to be associated with exosome or microvesicle localization from human plasma extracts (RNAlocate). 

We then performed enrichment analysis on functionally-validated targeted genes. Interestingly, GO enrichment analysis on cellular components indicated a strong enrichment of target genes located on endosomes and the Golgi membrane in both FC and ccEV extractions. Conversely, WP extracts showed a wider set of target genes that are part of pathways associated with several subcellular compartments (RISC, ribosome, nuclear and synaptic vesicles), while secEV was enriched for only a few targets that were associated with mitochondria and synaptic vesicles (Table 2).

### 3.4. FC and EV-Associated microRNA Profiles in Lung Cancer

EV-associated and free-circulating miRNA profiles were analyzed in a small cohort of patients with early-stage NSCLC (Table 1). Since the parallel extraction of EV-RNA and free-circulating RNA (ccEV and FC) produced enough good-quality sequencing data in healthy donors, we decided to extract and analyze only these fractions in the NSCLC cohort. Similar to healthy donors, we found differences in the sequencing performance between the two plasma fractions (EV and FC), with the FC extracts showing a higher percentage of PF reads and more miRNAs detected (Figure 5A). The average number of miRNAs identified in the NSCLC patients’ EV and FC extracts was consistent with the number of miRNAs detected in healthy donors. Specifically, in NSCLC cases a mean of 294 and 670 miRNAs were found in ccEV and FC extracts by at least 1 UMI, respectively (Figure 5A). Unsupervised analysis confirmed that ccEV and FC formed separate clusters in PCA, consistent with the data we obtained from the healthy donors. However, NSCLC patients and healthy donors showed distinct miRNA profiles (Figure 5B). This difference was further investigated by supervised analysis, which identified 31 miRNAs that were differentially expressed between NSCLC patients and healthy subjects in FC, and 29 miRNAs in ccEV (Appendix A). Then, we performed enrichment analysis of the top upregulated genes in NSCLC patients (16 for ccEV and 19 for FC), with a log2 fold change greater than two with respect to the healthy controls (Figure 5C). 

In contrast to healthy donors, both ccEV and FC miRNA fractions from lung cancer patients showed less enrichment for target genes associated with endosomes and Golgi membrane. Instead, GO enrichment indicated that fcmiRNAs and EV-associated miRNAs of lung cancer patients might have more target genes involved in other cellular processes, such as the formation of cytoplasmic stress granules and histone modifications, with respect to healthy donors (Table 3).

## 4. Discussion

The introduction of liquid biopsy for the identification of oncogenic biomarkers and for monitoring of patients’ responses to treatments has tremendously improved the routine of clinical practice for many diseases. However, so far FDA has approved the introduction of liquid biopsy assays only to detect alterations in plasma cell-free DNA, while detection and characterization of other circulating biomarkers, such as miRNAs or EVs, has been slow despite the technological advances, and the choice of the optimal method remains a challenge [14]. More than one study has compared the performances of different plasma RNA extraction methods and candidate biomarkers analysis [15,16,17,18], but only a few effectively analyzed the miRNA contents and miRNA sequencing performance between the different fractions of plasma [16,19,20].

Here, we studied the miRNA composition of different circulating fractions (total plasma, EV or free-circulating) by using alternative isolation protocols and compared their sequencing performance. In particular, we examined two different protocols for EV isolation from plasma: ccEV and secEV. The two techniques performed quite similarly, although the mean EV size and average number of EVs isolated from the same amount of plasma (2 mL) were slightly higher in secEV, possibly because we pooled multiple elution fractions together from the same chromatography. When analyzed independently by WB, the different secEV fractions showed increased expression of typical EV markers, from early elution to later elution fractions, indicating that SEC might actually comprise a wider range of extracellular particles, including EVs [21]. For example, F8-F11 did not express any of the EV markers analyzed, while F14–15 looked more specifically enriched in EVs. Instead, as is also reported in the literature, the kit we used to isolate EVs by CC (ccEV) should be suitable to isolate a few specific types of EVs, namely small EVs and exosomes [22,23]. Interestingly, we found some differences between secEV and ccEV in the expression of specific vesicular membrane markers, although most of these epitopes were detected at comparable levels. In both secEV and ccEV, characteristic tetraspanins CD9 and CD63 showed similar levels of expression, while CD81 was mostly expressed in ccEV extracts (*p* = 0.018). As well, some other epitopes, such as CD20, HLA-DR, CD11c, and MCSP1 were absent in secEV extracts, while they were detectable in ccEV. This difference may reflect the selective isolation of discrete subpopulations of EVs, which are normally distinguished by both the differential expression of specific membrane epitopes and the size of the vesicles derived from a specific cell type [24]. For example, CD81 might be exclusively found on exosomes and microvesicles, suggesting specificity for CC kits to isolate these specific subpopulations of EVs [25,26,27,28].

Much evidence exists that the RNA cargo of EVs may influence gene expression of specific target recipient cells. In particular, differential analysis of miRNAs transported by EVs might provide important implications about the heterogeneity in EV function and composition [29]. In our experiments, the profiles of RNAs extracted from different plasma fractions were similar, with an extremely low yield overall, and predominantly characterized by the presence of small RNA fragments in accordance with the literature [30]. However, because of the low concentrations of total RNA, which is normally obtained from liquid biopsy, we were limited in the possibility of studying the full spectrum of RNA species present in our samples, and we cannot exclude a possible presence of longer RNAs (lncRNAs and mRNAs, for example). These results limited the types of analysis that could be performed on these samples. However, in addition to the targeted detection of specific miRNA with high sensitivity approaches, such as digital PCR, the recent development of low-input RNA sequencing techniques has allowed for more sensitive analysis of the many species of circulating RNAs; however, some limitations still remain. Specifically, the concentration of input RNA is extremely low on average, limiting the accuracy of sample quantification. This issue could be minimized by the homogenization of input volumes used during RNA extraction and library preparation steps, and the addition of spike-in controls. In our study, we decide to compare miRNA expression profiles between different miRNA fractions extracted from plasma, since miRNAs are in general highly stable and are widely studied as cancer biomarkers [5,6,7]. We decided to use a commercially available method for miRNA library generation, which allows us to work with low input of total RNA and has been reported as extremely specific for miRNA, with low incorporation of other RNA contaminants (products of RNA degradation or other unspecific short RNA fragments) during library preparation [15,18,31,32]. In our samples, the library kit performed differently depending on the plasma RNA fraction that was used as input, with FC being the one mostly enriched in reads passing filters and aligned miRNA (64% and 78%, respectively). Conversely, both WP and EV fractions showed lower specificity, with reads ranging in mirBase from 20% to 32%. Differences in sequencing performance, depending on the plasma circulating fraction that is being analyzed, have been reported in the literature. For example, similarly to what we obtained in this study, miRNA sequencing of both whole plasma and EV fractions is reported to result in a high percentage of reads (ranging from 30% to 50%) that are too short to pass the filtering process and actually map on the genome [16,19]. However, data on the percentage of reads that align with miRNAs are discordant. For example, Kloten et al. reported that EV extractions showed fewer miRNA reads compared to WP [16]. On the contrary, Alsop et al. and Schneegans et al. showed a higher number of miRNA reads derived from the EV extractions [19,20]. In our samples, WP and EV fractions had a similar percentage of reads mapping on miRNAs, while the total number of identified miRNAs was lower in EV with respect to WP. Interestingly, in the work of Kloten et al. it was shown that the extraction method used might have some influence on the quantity and quality of miRNAs identified, with phenol extractions performing worse than spin-column-based approaches [16]. A possible explanation for the fact that we observed better performance of the FC sequencing compared to the other fractions could be that free RNases in the plasma normally degrade longer RNA species, and the sample results are enriched with miRNAs, which are protected by specific RNA-binding proteins [10,15]. Conversely, the long RNAs that are also included inside the EVs are protected from the RNases and more RNA species can be potentially present in addition to miRNAs, with a consequent reduction of the specific signal detected from miRNAs. Thus, lower concentrations of miRNAs in WP and EV extracts with respect to other RNA species, and presumably, a higher additional presence of contaminants or degraded RNA could have lowered the alignment rate of these fractions on mirBase. In fact, these aspects could partially compromise the quality of library preparation during the adapter-ligation steps and limit the ability to include in the ligation process only miRNAs, as demonstrated by the different library profiles obtained (Appendix A). We recognize that the present study has some limitations due to the methods we used for sample processing. In particular, we intentionally chose a library preparation kit that specifically limited the analysis to the miRNA class and in doing so we have excluded other RNA species (other small non-coding RNAs or mRNAs, for example), which could still represent a valuable source of potential clinical biomarkers. 

The difference in sequencing performances between the circulating RNA fractions was reflected on the number of identifiable miRNAs, with FC being the one with more miRNAs detectable, while EV-associated miRNA fractions (both secEV and ccEV) had lower amounts of miRNAs. Therefore, this result is likely indicative of a biological difference in the EV vs. non-EV miRNA plasma content and could be unrelated to the sequencing performance. We can speculate that, since extracellular vesicles have an active role in cellular signaling, the miRNAs detected are the only ones necessary for the biological processes that were active at the moment of plasma extraction. The lower specificity for miRNAs of EV and WP compared to FC could be only partially compensated by an increase of sequencing depth. To test this hypothesis, we evaluated different sequencing depths but only a modest benefit in the number of identifiable miRNAs was seen (Figure 3E,F). In fact, the addition of UMI during library preparation reduces the amplification bias and allows the proper counting of the RNA molecules present in the input RNA, improving the robustness of the assay regardless of the sequencing depth. Overall, we believe that in the context of liquid biopsy 20–25 million reads per sample allows a sufficient representation of the mirnome present in plasma, although for the FC fraction we suggest a decrease in the sequencing depth to 10 million reads to reach an optimized ratio between sequencing cost and miRNA detection. 

About 36% of miRNAs were shared between the three RNA fractions tested (including both EV isolation methods), however, the mirnome profiles varied substantially between each fraction. In particular, FC and ccEV were enriched for miRNAs with miRNA-targeted genes functionally associated with the Golgi membrane and endosomes, while the bulk analysis of the WP showed enrichment of miRNA-targets associated with subcellular compartments such as RISC, P-body, ribosome, nuclear and synaptic vesicles. Extracting both FC and EV RNA fractions simultaneously from the same aliquot of plasma is particularly important to reduce experimental biases and reduce biological variability. Hence, if applied to clinics, this process could be relevant in detecting candidate cancer biomarkers. Therefore, we decided to test this combined method for miRNA extraction and sequencing protocol on an NSCLC patient cohort. Differences in sequencing statistics between FC and EV fractions that were found in healthy subjects were confirmed in cancer patients, showing generally that a higher amount (in absolute quantity) of miRNAs can be detected in FC compared to ccEV. Interestingly, comparative analysis against healthy subjects highlighted the presence of some specific miRNAs predominantly found in NSCLC patients in both FC and EV fractions. In particular, 8 miRNAs out of the 16 top-ranking miRNAs, which were enriched in patients’ EVs (mir-425-3p, mir-30b-5p, mir-181a-2-3p, mir-93-3p, miR-139-5p, miR-339-3p, miR-339-5p, and hsa-miR-196b-5p), in addition to 2 miRNAs out of the 19 top upregulated FC miRNAs (miR-331-3p and let-7f) are well known biomarkers associated with lung cancer [33,34,35,36,37,38,39]. 

A relevant aspect is the presence of EVs in circulation that could originate from many cellular compartments or organs, including tumors. For example, many of the most represented EVs markers in secEV are also platelet (PLT) markers (CD31, CD41b, CD42a and CD62P), and it has already been reported that most of the isolated EVs might derive from PLTs or megakaryocytes [40,41]. It is indeed well known that tumor-educated platelets are an important component in the circulation of NSCLC patients and could have important prognostic and predictive implications [42,43,44]. Hence, PLTs-derived EVs (and their miRNA content) could also have important clinical implications. 

Globally, based on the analysis performed on the healthy donor samples, we can say that the separate analysis of free-circulating miRNAs and EV-associated miRNAs produces improved sequencing performance in terms of read specificity, and could be more informative with regard to the bulk analysis on the whole plasma miRNA extract. In fact, its application on miRNA analysis of NSCLC patients’ blood samples allowed the identification of some specific miRNAs known to be associated with lung cancer, with the potentiality to be used as clinical biomarkers for further applications. It will be very interesting to study the role of miRNAs present in the different comparts in relation to NSCLC patients’ prognosis and/or response to therapy, in order to identify a non-invasive biomarker potentially usable in clinical practice. A study with this aim is currently ongoing at our institute.

In particular, we would like to add some final conclusions on which isolation procedure would be the most suitable for vesicular miRNA extraction and analysis. Spin-column chromatography kits, such as those based on a silicon carbide resin matrix that we have used in the present study, surely provide a fast (less than three hours for 10–15 samples) and reliable way for the isolation of intact EVs and vesicular RNA purification in one shot. Most importantly, by using this approach we were able to simultaneously purify from the same aliquot of plasma both the EV-RNA and the free-circulating RNA (non-EV-associated), which is particularly important for a comparative analysis of pathological biomarkers. This approach also allows us to concentrate isolated RNA into a flexible elution volume (50 µL), which is ideal for subsequent small RNA sequencing library preparation. On the other hand, EV isolation with size-exclusion column chromatography requires more experience and accuracy by the operator in setting up and equilibrating the column and selecting the correct elution fractions that contain those nanoparticles in the range of 50–200 nm (the so-called EV zone) while excluding contaminants (free plasma proteins, for example). For this reason, size-exclusion techniques could be subjected to a higher chance of technical errors and variability, although SEC can be coupled with an automatic fraction collector that would limit manual errors. Additionally, the volumetric flow of the elution fraction is variable, and more than one elution must be collected to have an adequate amount of EVs, increasing the volume of the final eluate up to 1–1.5 mL. Therefore, we think that spin-column-based chromatography EV isolation methods might represent the cheapest, most practical and feasible approach for routine laboratory use and clinical sample analysis.

## Figures and Tables

**Figure 1 pharmaceutics-14-02029-f001:**
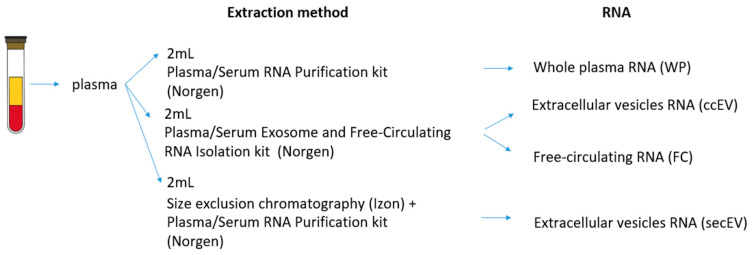
Schematic representation of the experimental procedure along with the different types of RNA extracted and analyzed from healthy donor’s plasma.

**Figure 2 pharmaceutics-14-02029-f002:**
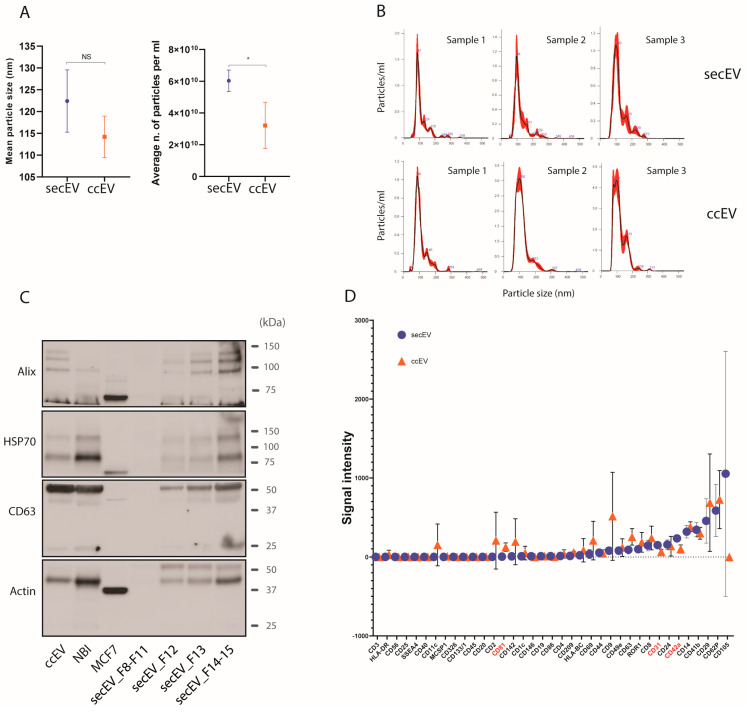
Evaluation of nanoparticles’ features in secEV and ccEV methods. (**A**) Boxplot representation of mean size and average plasma concentration (particles/mL) of EVs, as quantified at the NanoSight tracking system, comparing EVs extracted either with size exclusion chromatography (secCC) or column chromatography (ccEV), from three healthy donors; (**B**) Particle profile distribution, as detected by the NanoSight Software NTA for the EVs of the three samples analyzed, and obtained either by SEC (secEV) or CC (ccEV). For the secEV isolation the NTA profile represents the fraction with the highest concentration of EVs; (**C**) Representative western blot of main EV markers as detected on nitrocellulose membrane, following nanoparticles isolation from plasma by SEC (different elution fractions, from 8 to 15, are shown for SEC) and CC, and compared with a reference EV-isolation method (NBI, nickel-based isolation [13]) and a whole cell lysate of MCF7 cell line; (**D**) Average expression of characteristic membrane epitopes present on the EV surface, comparing vesicles extracted either with secEV or ccEV from healthy subjects (*n* = 3), as quantified by flow cytometry with the MACSPlex Exosome Kit, is indicated as Mean Fluorescence Intensity (MFI). In blue, the intensity values obtained from the secEV method, and in orange the values of ccEV (values have been normalized to blank controls). Markers that show a statistically significant difference in the expression between the two extraction methods are highlighted in red (single *p* values are reported in Appendix A). EV, extracellular vesicles; *, *p* < 0.05 and NS, not significant (two-way unpaired *t*-test).

**Figure 3 pharmaceutics-14-02029-f003:**
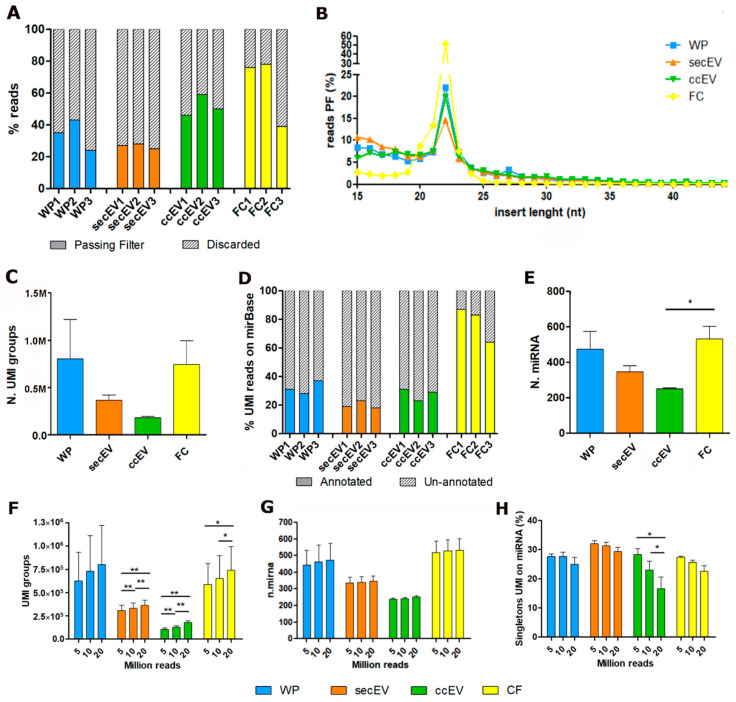
Comparison of sequencing performance according to extraction method. (**A**) Percentage of reads passing filter. (**B**) Length of inserts of passing filter reads. (**C**) Number of unique UMI groups identified. (**D**) Percentage of UMI-reduced reads mapping on miRBase v22. (**E**) Number of miRNAs identified, covered by at least 1 UMI. (**F**–**H**) Modulation of sequencing performance was checked according to the reduction in the number of reads (20 million vs. 10 million vs. 5 million reads) used for the analysis. (**F**) Number of unique UMI groups identified. (**G**) Number of miRNAs identified, covered by at least 1 UMI. (**H**) Percentage of miRNA called by only one UMI (singletons). *, *p* < 0.05; **, *p* < 0.01 (two-way unpaired *t*-test).

**Figure 4 pharmaceutics-14-02029-f004:**
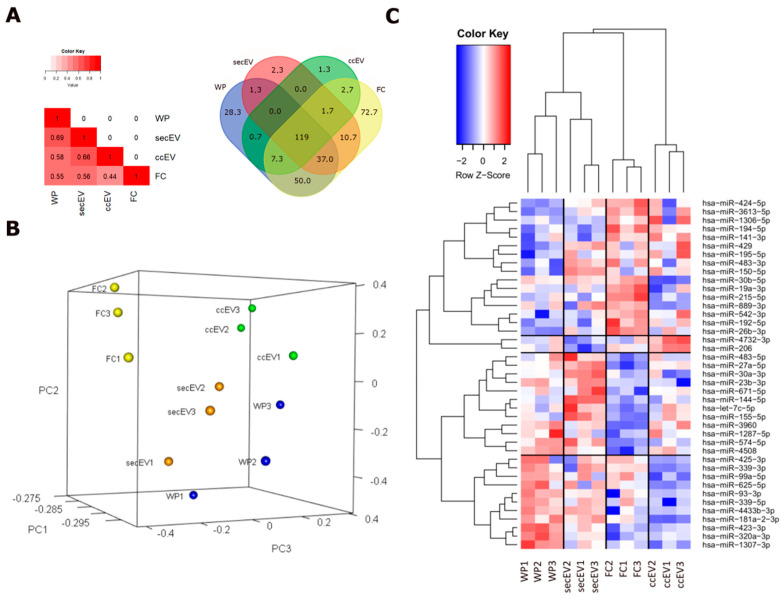
Differential miRNA expression analysis of the four extraction methods. (**A**) VENN diagram showing the number of identified miRNAs that are shared by one or more extractions. Number refers to the mean values obtained from the three healthy donors’ samples, considering those miRNAs covered by at least 5 UMI. (**B**) Unsupervised PCA analysis of miRNA. (**C**) Heatmap showing the top overexpressed miRNA in each extraction. Differential miRNA with Log2FC > |2| with an FDR < 0.05 are shown.

**Figure 5 pharmaceutics-14-02029-f005:**
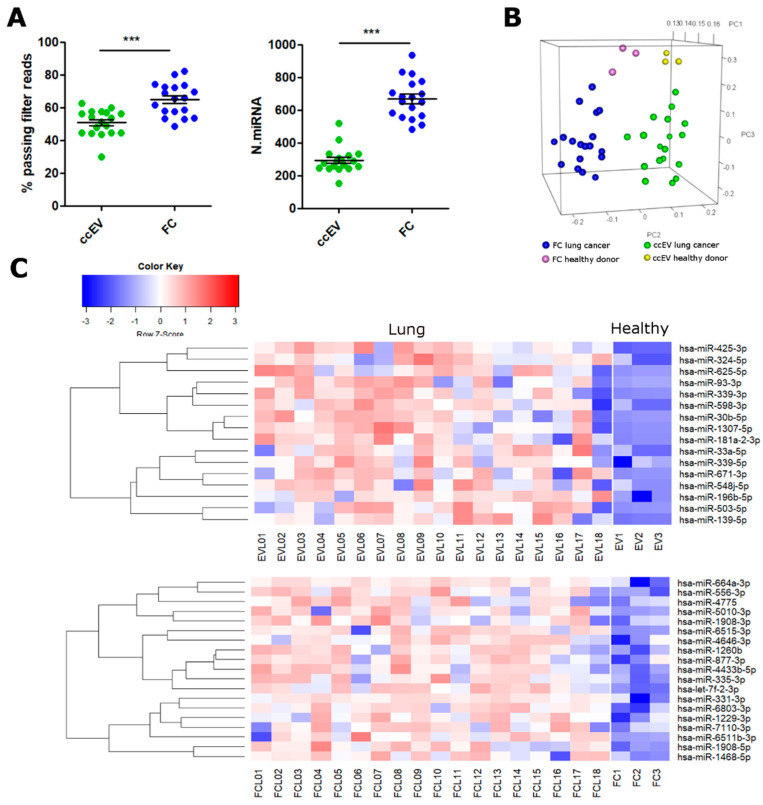
miRNA expression analysis in lung cancer plasma samples. (**A**) Sequencing performance of ccEV and FC extracts evaluated in 18 lung cancer patients’ plasma. Percentage of passing filter reads (left panel) and the number of miRNAs identified by at least 1 UMI (right panel) are shown. (**B**) PCA unsupervised analysis of lung cancer patients’ and healthy donors’ plasma miRNA. (**C**) Heatmap showing the top overexpressed miRNA in each extraction, in comparison with healthy donors’ plasma. Differential miRNA with Log2FC > |2| with an FDR < 0.05 are shown. ***, *p* < 0.001 (two-way unpaired *t*-test).

**Table 1 pharmaceutics-14-02029-t001:** Patients’ clinical characteristics.

Sample	Gender	Age	Histological Diagnosis ^1^	Stage	Smoking History
**L01**	M	72	ADC	IB	yes
**L02**	F	75	ADC	IIA	yes
**L03**	F	71	ADC	IIIA	yes
**L04**	F	68	ADC	IA	yes
**L05**	M	70	ADC	IIB	yes
**L06**	M	71	SQC	IIIA	yes
**L07**	F	72	ADC	IIIA	yes
**L08**	M	77	SQC	IB	yes
**L09**	M	55	ADC	IA	yes
**L10**	F	66	ADC	IB	no
**L11**	M	79	SQC	IIIA	yes
**L12**	F	70	SQC	IIB	yes
**L13**	F	55	ADC	IA	yes
**L14**	M	83	ADC	IB	no
**L15**	M	55	SQC	IIIA	yes
**L16**	M	72	ADC	IB	yes
**L17**	M	69	ADC	IIIA	yes
**L18**	M	72	ADC	IIIA	yes

^1^ ADC, adenocarcinoma; SQC, squamous cell carcinoma.

**Table 2 pharmaceutics-14-02029-t002:** Top cellular component enrichment results of each fraction, based on gene ontology analysis of the genes targeted by the miRNA found upregulated in the four extraction methods. Results with adjusted *p* value (BH) < 0.05 are reported.

RNA Fraction	Name	Hits	Pop Hits	*p* Value	Adj. *p* Value
WP	GO:0005844_polysome	12	42	0.0	0.0
WP	GO:0030122_AP-2_adaptor_complex	10	19	0.0	0.0
WP	GO:0032587_ruffle_membrane	18	105	0.0001	0.0056
WP	GO:0000792_heterochromatin	11	44	0.0002	0.0084
WP	GO:0005845_mRNA_cap_binding_complex	6	14	0.0005	0.0092
WP	GO:0005925_focal_adhesion	50	498	0.0004	0.0092
WP	GO:0008021_synaptic_vesicle	20	134	0.0003	0.0092
WP	GO:0016442_RISC_complex	7	20	0.0005	0.0092
WP	GO:0016581_NuRD_complex	7	21	0.0006	0.0092
WP	GO:0016607_nuclear_speck	45	443	0.0006	0.0092
secEV	GO:0005758_mitochondrial_intermembrane	12	98	0.0003	0.0127
secEV	GO:0005759_mitochondrial_matrix	31	447	0.0004	0.0127
secEV	GO:0014069_postsynaptic_density	22	270	0.0003	0.0127
secEV	GO:0005942_PI3K_complex	5	25	0.0026	0.0494
secEV	GO:0043235_receptor_complex	19	257	0.0022	0.0494
ccEV	GO:0005802_trans-Golgi_network	9	207	0.0008	0.0132
ccEV	GO:0055037_recycling_endosome	8	146	0.0004	0.0132
ccEV	GO:0005770_late_endosome	7	157	0.0026	0.0214
ccEV	GO:0005925_focal_adhesion	14	498	0.0024	0.0214
ccEV	GO:0005776_autophagosome	5	91	0.0044	0.029
ccEV	GO:0000139_Golgi_membrane	16	682	0.0066	0.0311
ccEV	GO:0030667_secretory_granule_membrane	5	100	0.0064	0.0311
ccEV	GO:0005769_early_endosome	9	301	0.0087	0.0359
FC	GO:0005769_early_endosome	31	301	0.0	0.0
FC	GO:0031901_early_endosome_membrane	20	173	0.0	0.0
FC	GO:0071141_SMAD_protein_complex	5	8	0.0	0.0
FC	GO:1990124_messenger_ribonucleoprotein_comp	5	11	0.0001	0.0025
FC	GO:0000139_Golgi_membrane	44	682	0.0002	0.004
FC	GO:0016363_nuclear_matrix	12	112	0.001	0.0156
FC	GO:0055037_recycling_endosome	14	146	0.0011	0.0156
FC	GO:0030014_CCR4-NOT_complex	5	20	0.0013	0.0161
FC	GO:0035098_ESCE(Z)_complex	5	23	0.0022	0.0242
FC	GO:0005635_nuclear_envelope	17	217	0.0027	0.0264

**Table 3 pharmaceutics-14-02029-t003:** Top cellular component enrichment results in lung cancer patients, based on gene ontology analysis of the genes targeted by the miRNA found upregulated in FC and ccEV fractions. Results with adjusted pvalue (BH) < 0.05 are reported.

RNA Fraction	Name	Hits	Pop Hits	*p* Value	Adj. *p* Value
FC	GO:0000123_histone_acetyltransferase_complex	7	27	0.0003	0.0119
FC	GO:0030122_AP-2_adaptor_complex	6	19	0.0003	0.0119
FC	GO:0031209_SCAR_complex	6	17	0.0002	0.0119
FC	GO:0017146_NMDA_selective_glutamate_recept.	6	21	0.0005	0.0149
FC	GO:0032591_dendritic_spine_membrane	5	15	0.0008	0.0159
FC	GO:0090575_RNA_polymerase_II_transcription	11	80	0.0007	0.0159
FC	GO:0010494_cytoplasmic_stress_granule	11	84	0.0011	0.0187
FC	GO:0071782_endoplasmic_reticulum_tubular_nt	5	20	0.0023	0.0342
FC	GO:0098982_GABA-ergic_synapse	9	69	0.003	0.0397
FC	GO:0045211_postsynaptic_membrane	14	146	0.0037	0.044
ccEV	GO:0000159_protein_phosphatase_type2A_compl.	7	32	0.0	0.0
ccEV	GO:0032593_insulin-responsive_compartment	5	13	0.0	0.0
ccEV	GO:0010494_cytoplasmic_stress_granule	9	84	0.03	1.11
ccEV	GO:0032991_protein-containing_complex	32	679	0.04	1.11
ccEV	GO:0000792_heterochromatin	6	44	0.09	2.08
ccEV	GO:0036464_cytoplasmic_ribonucleoprot_granule	8	85	0.13	2.34
ccEV	GO:0035097_histone_methyltransferase_complex	5	33	0.16	2.42
ccEV	GO:0000307_CDK_holoenzyme_complex	5	35	0.21	3.06
ccEV	GO:0000932_P-body	9	116	0.24	3.09
ccEV	GO:0030027_lamellipodium	12	198	0.36	4.16

## Data Availability

The data presented in this study are available on request from the corresponding author.

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
