# Peer review of "Comparative Analysis of Free-Circulating and Vesicle-Associated Plasma microRNAs of Healthy Controls and Early-Stage Lung Cancer Patients"

_pharmaceutics, 2022, doi:10.3390/pharmaceutics14102029_

Round 1

Reviewer 1 Report

In this research article, Pasini et al. tested whether high heterogeneity in sample processing, isolation of circulating fraction, RNA extraction and sequencing, affects the reproducibility and the introduction of circulating biomarkers in clinical practice. Thus, they performed a comparative analysis using multiple methodologies of miRNAs in three different fractions of healthy donors and early stage lung cancer patient blood: whole plasma (WP), free-circulating (FC) and intra-extracellular vesicles (EV) (isolated by column (ccEV) or size exclusion chromatography (secEV)). In parallel, they also compare the protein profile of ccEV and secEV.

They demonstrated that sequencing performance and the mirnome profile varied between RNA extracted with different isolation methods with better sequencing performance in FC samples. Higher varieties of miRNAs were identified in WP and FC with respect to EV independently on the methods used to isolated EV. Importantly, the analysis of free circulating and intra-EV miRNA profiles in lung cancer patients demonstrated the reliability of the biomarkers identifiable on plasma with these approaches. In addition, they found slight differences between ccEV and secEV in expression of EV markers.

The paper is well written and flows logically. However, important considerations regarding mostly related to EV field are missing.

Main comments:

-The EV are not fully characterized in accordance with the guidelines MISEV2018 regarding the nomenclature. Why did the Authors decide to use the general term “extracellular vesicles”?

-Why did the Authors use column and size exclusion chromatography to isolate EV? Why did not they use other methods as well?

-In figure 2, why is there breast cancer MCF7 lysate and not lung cell lysate and lung cancer cell lysate (adenocarcinoma and squamous carcinoma cell lines)?

-To isolate the microRNAs encapsulated into the EV, RNAse digestion would be suggested to eliminate any external RNA attached to the EV membrane

-EV isolated from lung cancer patients should be characterized (NTA, markers)

-Are there quantitative differences of microRNAs in ccEV or FC in cancer patients compare donors?

Reviewer 2 Report

This manuscript (pharmaceutics-1871992) compared the miRNA sequencing content in two different isolation protocols of extracellular vesicles (EVs), in free-circulating (FC) fraction and in whole plasma (WP) from three healthy volunteering donors. Furthermore, the authors chosen the parallel extraction of EV-RNA and FC RNA to analyse the miRNAs in a non-small cell lung cancer (NSCLC) patient cohort. Overall, this is a technically well-conducted study, reasonably well written, with an interesting topic. In fact, this type of comparative studies are quite interesting and contribute to the potential development of feasible biomarkers for clinical use. However, in this manuscript, I think that the authors did not fully explore the data they obtained. Some commentaries and concerns:

1.       This manuscript seems a mix of two works. The final part, where the authors evaluate the NSCLC patients did not relate with the first part, which is a classic comparative study. Furthermore, the authors did not clearly elucidate what conclusions they reach in comparing the two different isolation protocols of EVs. Which one is the best to be used for sequencing miRNAs? And for clinical use? Which is the cheapest? And the most feasible?

2.       The NSCLC cohort was little explored. As the authors showed in Table 1, there were patients with different histological diagnosis (adenocarcinoma or squamous cell carcinoma) and different cancer stage. Why were the patients all grouped together? The authors should stratify the patients by histological diagnosis and cancer stage because patients with different disease severity degrees, probably have different pools of released miRNAs.

3.       In comparative part, the authors focused on the 36% of identified miRNAs shared between the four RNA extraction methods. However, it would be more interesting whether the authors enumerate the miRNA(s) that were differently expressed among the different extraction methods, mainly between the FC and intra-EV content, since it is already described that there are miRNAs with different profiles between these two fractions of plasma. Moreover, are there common miRNAs in the secEV and ccEV groups that do not exist in FC?

4.       The authors enumerated several other comparative studies. However, they did not discuss their conclusions, neither show the way how the results presented in this manuscript could relate to them.

5.       To achieve a conclusion of the study of NSCLC patient cohort, it is fundamental suggest, evaluate and validate a miRNA or a miRNAs panel with biomarker potential to distinguish sensitively and specifically NSCLC patients from healthy volunteering donors.

6.       The discussion section was almost all referring to the comparative study with samples from three healthy volunteering donors. The authors should explore more the results obtained from NSCLC patients, discussing their potential as clinical biomarkers.

Minor concerns:

1.       Some problems with the definition of abbreviations at first usage - whole plasma (line 69); free-circulating (line 50 and 197); column chromatography (ccEV) (set twice)

2.       In VENN diagram (Fig. 4a), what was the meaning of MV and EV?

Reviewer 3 Report

In the MS by Pasini et al., the Authors perform miRNAs sequencing on plasma-derived samples, highlighting the strengths and limitations of this approach, comparing the results obtained from whole plasma (WP) miRNA, cell free miRNA (FC), and miRNA contained in extracellular vesicles (EVs). They tested side by side the isolation of blood EVs with column chromatography (ccEV) or with size exclusion chromatography (secEV) and they verified the obtained data on a small cohort of early-stage NSCLC patients.

The study is well conceived, the topic is relevant, and the experimental part is well performed nonetheless two main aspects limit my enthusiasm for the reported results:

1. Even if the methods used to characterize the EV in the samples derived from healthy donors are coherent, in my opinion most of the isolated EVs derive from platelets (PLTs) or megacariocytes as already reported (Flaumenhaft R et al. 2010, PMID 21049389; Italiano et al., 2010, PMID 20739880). This is, also, attested by the fact that many of the most represented EVs markers in secEVs are also PLTs markers (see Table S1: CD31, CD41b, CD42a, CD62P). Generally speaking, this is one of the most important limitations to apply EV-associated miRNAs analysis to diagnostic protocols. In this case, how and why miRNAs derived from the main part from PLTs could vary in NSCLC? Authors should at least discuss these aspects in the Discussion section.

2. Authors show that RNAs isolated from all sample types were in the range of 50-100 nts (cfr. Fig. S1). This is unexpected for me because many authors report the presence of longer RNAs (lncRNAs and even mRNAs) into EVs. As the kits the authors used to isolate RNAs (i.e. Norgen Kits) are able to isolate all type of RNAs and the kit they used to quantify RNAs (i.e. RNA6000 pico chips) is able to detect larger RNAs my question is: are the diagrams in Fig. S1A referred only on miRNAs range or, as it seems, the samples contain only small size RNAs. In the latter case how the authors explain this unexpected result? This point is also coherent with the result shown in Fig. 3A: reads from FC samples which is expected, indeed, almost formed by miRNAs, the only RNA class protected by serum RNases, gave the mostly enriched in reads passing filters and aligned miRNAs. Authors discuss this point by hypothesizing the presence of contaminants in WP and EV extracts (lines 422-424), but in my opinion these “contaminats” are rather other RNAs (mRNA? lncRNAs? rRNAs?) than non only should not considered as “contaminants” as authors did, but they could also be more specific than miRNAs from the diagnostic point of view (for ex mRNAs coding for tumor-associated or tumor-specifc Ags). Authors should discuss also this point in Discussion section, highlighting these bias and limitations.

Minor points:

1. Some extra spaces are present throughout the MS

2. Line 379: authors report that 1ml of plasma was used to isolate EVs but throughout the MS it was reported 2 ml (See also Fig. 1). Authors should clarify.

3. Line 461-462: “In particular, 8 out 16 top-ranking miRNAs enriched in patients’ the EVs” should be revised.
